# Association between pre-existing Pulmonary Hypertension and COVID-19 related outcomes in inpatient and ambulatory care settings

**Shilpa Vijayakumar**[1], **David W. Louis**[2], **Emily Corneau**[3], **Sebhat Erqou**[3,4,5], **Stephen W. Waldo**[6,7,8], **Mary E. Plomondon**[6], **Madhura Gokhale**[6], **Wasiq Sheikh**[3,4,5], **Phinnara Has**[9], **Saad Marwan**[4,5], **J. Dawn Abbott**[4,5], **Matthew Jankowich**[3,10], **Herbert D. Aronow**[11,12], **Wen-Chih Wu**[3,4,5], **Gaurav Choudhary**[3,4,5*]

**1** Cardiovascular Imaging Program, Cardiovascular Division and Department of Radiology, Brigham and Women's Hospital and Harvard Medical School, Boston, Massachusetts, United States of America, **2** Division of Cardiology, Primed Cardiology, Northeast Medical Group, Trumbull, Connecticut, United States of America, **3** Providence Veterans Affairs Medical Center, Providence, Rhode Island, United States of America, **4** Department of Medicine, Division of Cardiology, Alpert Medical School of Brown University, Providence, Rhode Island, United States of America, **5** Lifespan Cardiovascular Institute, Providence, Rhode Island, United States of America, **6** CART Program, Office of Quality and Patient Safety, Veterans Health Administration, Washington DC, United States of America, **7** Rocky Mountain Regional VA Medical Center, Aurora, Colorado, United States of America, **8** University of Colorado School of Medicine, Aurora, Colorado, United States of America, **9** Lifespan Biostatistics, Epidemiology, and Research Design (BERD) Core, Providence, Rhode Island, United States of America, **10** Division of Pulmonary, Critical Care and Sleep Medicine, Alpert Medical School of Brown University, Providence, Rhode Island, United States of America, **11** Heart & Vascular Services, Henry Ford Health, Detroit, Michigan, United States of America, **12** Michigan State University College of Human Medicine, East Lansing, Michigan, United States of America

* gaurav_choudhary@brown.edu

## Abstract

### Background

Afflicting up to 1% of population, pulmonary hypertension (PH) is commonly associated with cardiopulmonary and metabolic diseases, but the effect of COVID-19 in patients with pre-existing PH remains unclear.

### Methods

We conducted a retrospective cohort study in patients who had undergone right-heart-catheterization within the VA Healthcare system and had a subsequent hospital admission with COVID-19 (*inpatient cohort, n=1204*) or had COVID-19 positivity but not admitted (*outpatient cohort, n=6576*). Inpatient findings were confirmed in a non-VA *validation cohort (n=656)* who had undergone echocardiography with subsequent admission. PH was defined invasively as mean pulmonary artery pressure (mPAP) >20 mmHg and non-invasively as estimated right ventricular systolic pressure (RVSP) >30 mmHg. In-hospital outcomes (inpatient cohort) and 1-year mortality (outpatient cohort) were assessed using multivariable logistic or Cox regression adjusting for confounders.

**Data availability statement:** All relevant data are within the manuscript and its Supporting Information files.

**Funding:** The author(s) received no specific funding for this work.

**Competing interests:** NO authors have competing interests

## Results

Pre-existing PH was independently associated with greater in-hospital mortality (PH using mPAP: adjusted odds ratio [aOR] 1.60, 95%CI: 1.04–2.46; PH using RVSP: aOR 2.12, 95% CI 1.18–3.82). Among outpatients, those with COVID-19 had >8-fold higher 90-day and 2.8 fold higher 91–365 day adjusted hazard of mortality irrespective of PH status. Hazards of 90-day hospitalization were similarly driven by COVID-19. The findings were comparable for patient subgroup with normal pulmonary capillary wedge pressures.

## Conclusion

Pre-existing PH is independently associated with higher in-hospital COVID-19 mortality. In outpatients, COVID-19 positivity was associated with increased mortality over 1 year irrespective of PH status, with highest risk within the first 90 days.

## Introduction

Pulmonary hypertension (PH) is defined as a mean pulmonary artery pressure (mPAP) > 20 mmHg during right heart catheterization (RHC) [1,2]. PH results from multifactorial causes, encompassing both pre-capillary and post-capillary, ultimately leading to right ventricular (RV) overload and dysfunction [2]. Its global prevalence is significant, affecting up to 1% of the population, often associated with older age and cardiopulmonary and metabolic comorbidities [1]. Approximately 80% of patients undergoing RHC have PH, and nearly 40% of all clinically conducted echocardiograms reveal elevated pulmonary artery systolic pressure (PASP), suggestive of PH [1]. Five-year mortality among patients with PH ranges from 25 to 40%, regardless of diagnostic method (invasive hemodynamics or echocardiography) [1].

Coronavirus disease-2019 (COVID-19), caused by severe acute respiratory syndrome-coronavirus-2 (SARS-CoV-2), has affected over 700 million people worldwide, resulting in over 6.9 million deaths [3]. Poor COVID-19 outcomes are significantly associated with factors such as older age, systemic hypertension, cardiovascular disease, and pre-existing lung and metabolic diseases [4,5]. Acute COVID-19 infection is also associated with pulmonary embolism and right heart failure due to endothelial cell dysfunction, hypercoagulability, and micro- and macro-vascular thrombosis [6,7], possibly attributed to SARS-CoV-2's particular affinity for pulmonary vasculature [8]. Single-center observational data suggest that COVID-19 can lead to acute PH in patients admitted to either non-critical or intensive care units, a finding which is associated with an increased risk of death [9,10]. However, the association between pre-existing PH and COVID-19 outcomes remains unclear.

Understanding the relationship between pre-existing PH and COVID-19 outcome, both acute and long-term mortality, may identify a high-risk cohort of patients requiring aggressive short-term intervention such as COVID-19 preventive therapy and treatment, and closer long-term monitoring and cardiopulmonary risk mitigation after recovery from acute COVID-19 infection.

In the present study, we assess the association between pre-existing PH (identified through RHC or echocardiogram) and COVID-19 outcomes in hospitalized patients. Additionally, we explore the association between pre-existing PH and long-term mortality in patients with positive outpatient COVID-19 testing [11].

## Methods

### Data source

A retrospective analysis of a nationwide cohort of Veterans was performed using the Clinical Assessment, Reporting and Tracking (CART) Program data from the Department of Veterans Affairs [12,13], the largest pulmonary hemodynamics database available to date, linked to the VA enrollment files, inpatient and outpatient encounters. This study was supported using data from the VA COVID-19 Shared Data Resource. This study was approved by the Institutional Review Board of the Providence VA Medical Center.

### Study population

VHA-enrolled Veterans ≥ 18 years of age who underwent RHC with mPAP between January 1, 2016, and December 31, 2019, and who subsequently received a COVID-19 test (PCR or antigen) within the VA system between March 1, 2020, and August 30, 2022 were included. We further stratified the sample based on inpatient vs. outpatient test setting. The inpatient setting was defined as testing positive for COVID-19 four days prior to admission to a VA medical center and/or up to 48 hours after admission, defining the *inpatient cohort*. The outpatient setting included Veterans who never tested positive in an inpatient setting per the above definition who received a COVID-19 test at a VA outpatient lab, defining the *outpatient cohort*. We took the Veteran's first test and assigned the Veteran as COVID-19 positive according to the result. If the Veteran's first test was negative, we followed them until the end of the study period or censored them upon testing positive. For COVID-19 positive Veterans, we followed them from the first positive test until the end of the study period, regardless of whether they tested negative previously. Therefore, the group of patients who were initially COVID-19 negative but subsequently became COVID-19 positive were included in both COVID-19 negative and COVID-19 positive groups with different follow-up periods depending on when they turned positive.

Our *validation cohort* included patients ≥ 18 years of age admitted between March 1– December 31, 2020 with COVID-19 diagnosis in three hospitals in Rhode Island (the Rhode Island, Miriam, and Newport Hospitals) with an available prior echocardiogram with estimation of right ventricular systolic pressure (RVSP).

Flow charts for selection of each cohort are shown in **Supplement 1** in S1 Data.

Baseline demographics, comorbidities, vaccination status, smoking status, date of COVID-19 testing, and clinical data were collected. Comorbidities were identified from the electronic health record using ICD-10 diagnosis codes. The Charlson Comorbidity Index (CCI) was calculated for each patient using ICD-10 codes [14].

### Exposure

For the *inpatient* and *outpatient cohorts,* the exposure of interest was invasively measured PH, defined as a resting mPAP > 20 mmHg, as obtained by RHC. For the *validation cohort*, PH was defined as RVSP > 30 mmHg, chosen due to data showing comparable outcomes between RVSP > 30 mmHg and mPAP > 20 mmHg [1]. The exposure was modeled separately as both a binary (PH, mPAP >20/RVSP >30 mmHg or no PH, mPAP≤ 20/RVSP≤ 30 mmHg) and continuous variable.

### Outcomes

For the *inpatient* and *validation cohorts,* the primary outcome of interest was in-hospital all-cause mortality. The *validation cohort* was only used for validation of the primary outcome. For the *inpatient cohort*, secondary endpoints included

intensive care unit level of care during hospitalization, mortality within 30 days of admission, mechanical ventilation within 30 days of admission, vasopressor medication use within 30 days of admission, high-flow oxygen use within 30 days of admission, low-flow oxygen use within 30 days of admission, and dialysis use within 30 days of admission. For the *outpatient cohort*, the primary outcome of interest was 1-year all-cause mortality and secondary outcome was 1-year all-cause hospitalization. Dates of death were confirmed using VA's Vital Status File and Master Patient Index, which contain mortality data from multiple VA and non-VA data sources with a validated 98.3% sensitivity and 97.6% agreement with the National Death Index.

## Statistical analyses

### Inpatient and validation cohorts

For analysis of primary in-hospital mortality in the *inpatient* and *validation cohorts*, we estimated odds ratios of in-hospital mortality using logistic-regression models adjusting for demographic and a wide range of potential clinical confounders (Table 4 footnote) with VA facilities included as fixed effects. Secondary outcomes mentioned above were analyzed in a similar fashion. In a subgroup analysis, we also stratified the relationship between mPAP and in-hospital mortality by pulmonary capillary wedge pressure (PCWP) > 15 mmHg or ≤ 15 mmHg to assess the impact of elevated left-sided filling pressures on outcome.

### Outpatient cohort

For analysis of 1-year all-cause mortality, we compared patient characteristics and generated Kaplan-Meier survival curves by PH and COVID-19 test result starting at day 0 of COVID-19 positivity, thus comparing four groups: negative outpatient COVID-19 test and mPAP ≤ 20 mmHg, negative outpatient COVID-19 test and mPAP > 20 mmHg, positive outpatient COVID-19 test and mPAP ≤ 20 mmHg, and positive outpatient COVID-19 test and mPAP > 20 mmHg. Our primary analysis of 1-year mortality analysis violated the proportional hazard assumption and thus, we used a two-period analysis (0–90 day mortality, and 91–365 day mortality) based on the crossover time of the mortality curves (survival probability curve in **Supplement 2** in S1 Data).

We estimated hazard ratios using Cox-regression models adjusting for demographic and a wide range of potential clinical confounders (Table 6 footnote) with VA facilities included as fixed effects.

All analyses were performed using SAS Enterprise Guide 7.1 (SAS Institute Inc, Cary, NC). P-value <0.05 was considered statistically significant, and p-value <0.0167 was considered significant for subgroup analyses to account for multiple testing.

## Results

### Baseline characteristics

Baseline demographic and clinical characteristics for the *inpatient cohort*, stratified by mPAP, are shown in **Table 1**. A total of 1204 Veterans were included, with mean age (SD) 72.1 (9.6), 96.9% male, 65.5% White and 28.8% Black. Of these patients, 252 (21%) had a normal mPAP of ≤ 20 mmHg and 952 (79%) had PH with mPAP > 20 mmHg. Compared to those without PH, patients with PH were younger, had higher body mass index (BMI), and were more likely to have comorbidities such as congestive heart failure (CHF), diabetes mellitus, obstructive sleep apnea (OSA), and chronic obstructive pulmonary disease (COPD). Baseline medications are presented in **Supplement 3** in S1 Data.

Baseline characteristics, stratified by RVSP, in the *validation cohort* are presented in **Supplement 4** in S1 Data. A total of 656 patients were included, of which 395 (60%) had an RVSP of ≥ 30 mmHg and 261 (40%) had an RVSP < 30 mmHg.

Baseline demographic and clinical characteristics, in the *outpatient* cohort, stratified by mPAP and by COVID-19 outpatient test positivity, are shown in **Table 2**. Patients with PH were more likely to have comorbidities including myocardial infarction, CHF, hypertension, diabetes, chronic kidney disease, and COPD compared to patients without PH.

**Table 1. Baseline Characteristics for Analysis of Inpatient Cohort.**

| | Total (n = 1204) | mPAP ≤ 20 mmHg (n = 252) | mPAP > 20 mmHg (n = 952) | p-value |
|---|---|---|---|---|
| Age (years), mean (SD) | 72.1 (9.6) | 73.0 (10.7) | 71.8 (9.3) | 0.09 |
| BMI (kg/m²), mean (SD) | 30.0 (7.4) | 28.7 (7.1) | 30.3 (7.4) | 0.002 |
| Male sex | 1167 (96.9%) | 246 (97.6%) | 921 (96.7%) | 0.47 |
| Race | | | | |
| American Indian or Alaska Native | 5 (0.4%) | 2 (0.8%) | 3 (0.3%) | 0.002 |
| Asian | 3 (0.2%) | 2 (0.8%) | 1 (0.1%) | |
| Black or African American | 347 (28.8%) | 56 (22.2%) | 291 (30.6%) | |
| Native Hawaiian/Other Pacific Islander | 2 (0.2%) | 2 (0.8%) | 0 (0.0%) | |
| Unknown | 58 (4.8%) | 14 (5.6%) | 44 (4.6%) | |
| White | 789 (65.5%) | 176 (69.8%) | 613 (64.4%) | |
| Vaccination Status | | | | |
| None | 715 (59.4%) | 146 (57.9%) | 569 (59.8%) | 0.035 |
| One dose | 309 (25.7%) | 56 (22.2%) | 253 (26.6%) | |
| Two or more doses | 180 (15.0%) | 50 (19.8%) | 130 (13.7%) | |
| RHC within 1 year of positive test | 171 (14.2%) | 130 (11.9%) | 141 (14.8%) | 0.24 |
| Date of COVID Test | | | | |
| 3/1/2020-8/30/2020 | 147 (12.2%) | 28 (11.1%) | 119 (12.5%) | 0.81 |
| 9/1/2020-2/28/2021 | 379 (31.5%) | 74 (29.4%) | 305 (32.0%) | |
| 3/1/2021-8/30/2021 | 153 (12.7%) | 36 (14.3%) | 117 (12.3%) | |
| 9/1/2021-2/28/2022 | 379 (31.5%) | 82 (32.5%) | 297 (31.2%) | |
| 3/1/2022-8/30/2022 | 146 (12.1%) | 32 (12.7%) | 114 (12.0%) | |
| Smoking Status | | | | |
| Current Smoker | 130 (10.8%) | 21 (8.3%) | 109 (11.4%) | 0.45 |
| Former Smoker | 641 (53.2%) | 143 (56.7%) | 498 (52.3%) | |
| Never Smoker | 375 (31.1%) | 76 (30.2%) | 299 (31.4%) | |
| Unknown | 58 (4.8%) | 12 (4.8%) | 46 (4.8%) | |
| Comorbidities* | | | | |
| Prior Myocardial Infarction | 173 (14.4%) | 41 (16.3%) | 132 (13.9%) | 0.33 |
| CHF | 901 (74.8%) | 146 (57.9%) | 755 (79.3%) | <0.001 |
| Hypertension | 1138 (94.5%) | 235 (93.3%) | 903 (94.9%) | 0.32 |
| Ischemic heart disease | 905 (75.2%) | 192 (76.2%) | 713 (74.9%) | 0.67 |
| Diabetes mellitus | 787 (65.4%) | 143 (56.7%) | 644 (67.6%) | <0.001 |
| Chronic kidney disease | 721 (59.9%) | 137 (54.4%) | 584 (61.3%) | 0.044 |
| Chronic liver disease | 192 (15.9%) | 31 (12.3%) | 161 (16.9%) | 0.075 |
| COPD | 582 (48.3%) | 96 (38.1%) | 486 (51.1%) | <0.001 |
| OSA | 665 (55.2%) | 121 (48.0%) | 544 (57.1%) | 0.010 |
| Cancer | 335 (27.8%) | 76 (30.2%) | 259 (27.2%) | 0.35 |

Continuous variables presented as mean (standard deviation) and analyzed using T-test. Dichotomous variables presented as frequency (percentage) and analyzed using Chi-square testing.

Abbreviations: BMI = body mass index; COPD = chronic obstructive pulmonary disease; CHF = congestive heart failure; OSA = obstructive sleep apnea; mPAP = mean pulmonary artery pressure; RHC = right heart catheterization

*All comorbidities defined by Charlson comorbidity definitions, determined using ICD10 codes from VA claims.

**Table 2.  Baseline Characteristics for Analysis of Outpatient Cohort.**

| | Negative Outpatient COVID Test, mPAP ≤ 20 mmHg (n=1766) | Positive Outpatient COVID Test, mPAP ≤ 20 mmHg (n=721) | Negative Outpatient COVID Test, mPAP > 20 mmHg (n=3829) | Positive Outpatient COVID Test, mPAP > 20 mmHg (n=1581) | p-value |
|---|---|---|---|---|---|
| Age (years), mean (SD) | 69.4 (10.5) | 69.8 (10.3) | 70.5 (9.3) | 70.6 (9.4) | <0.001 |
| BMI (kg/m$^2$), mean (SD) | 29.4 (5.7) | 29.7 (5.5) | 31.5 (6.9) | 31.8 (7.1) | <0.001 |
| Male sex | 1679 (95.1%) | 684 (94.9%) | 3622 (94.6%) | 1507 (95.3%) | 0.70 |
| Race | | | | | |
| White | 1380 (78.1%) | 559 (77.5%) | 2813 (73.5%) | 1169 (73.9%) | 0.002 |
| Black | 295 (16.7%) | 120 (16.6%) | 808 (21.1%) | 327 (20.7%) | |
| Other | 91 (5.2%) | 42 (5.8%) | 208 (5.4%) | 85 (5.4%) | |
| Vaccination Status | | | | | |
| None | 1143 (64.7%) | 308 (42.7%) | 2583 (67.5%) | 731 (46.2%) | <0.001 |
| One dose | 440 (24.9%) | 192 (26.6%) | 949 (24.8%) | 421 (26.6%) | |
| Two or more doses | 183 (10.4%) | 221 (30.7%) | 297 (7.8%) | 429 (27.1%) | |
| RHC within 1 year of positive test | 140 (7.9%) | 83 (11.5%) | 289 (7.5%) | 169 (10.7%) | <0.001 |
| Date of COVID Test | | | | | |
| 3/1/2020-8/30/2020 | 472 (26.7%) | 58 (8.0%) | 1042 (27.2%) | 155 (9.8%) | <0.001 |
| 9/1/2020-2/28/2021 | 587 (33.2%) | 163 (22.6%) | 1321 (34.5%) | 403 (25.5%) | |
| 3/1/2021-8/30/2021 | 323 (18.3%) | 63 (8.7%) | 776 (20.3%) | 141 (8.9%) | |
| 9/1/2021-2/28/2022 | 241 (13.6%) | 295 (40.9%) | 475 (12.4%) | 554 (35.0%) | |
| 3/1/2022-8/30/2022 | 143 (8.1%) | 142 (19.7%) | 215 (5.6%) | 328 (20.7%) | |
| Smoking Status | | | | | |
| Current or Former Smoker | 1118 (63.3%) | 418 (58.0%) | 2468 (64.5%) | 975 (61.7%) | 0.006 |
| Comorbidities* | | | | | |
| Myocardial Infarction | 88 (5.0%) | 46 (6.4%) | 263 (6.9%) | 127 (8.0%) | 0.004 |
| CHF | 761 (43.1%) | 337 (46.7%) | 2348 (61.3%) | 1014 (64.1%) | <0.001 |
| Hypertension | 1522 (86.2%) | 640 (88.8%) | 3485 (91.0%) | 1454 (92.0%) | <0.001 |
| Ischemic Heart Disease | 1153 (65.3%) | 473 (65.6%) | 2606 (68.1%) | 1083 (68.5%) | 0.10 |
| Diabetes | 791 (44.8%) | 358 (49.7%) | 2192 (57.2%) | 968 (61.2%) | <0.001 |
| Chronic Kidney Disease | 490 (27.7%) | 244 (33.8%) | 1443 (37.7%) | 678 (42.9%) | <0.001 |
| Liver Disease | 189 (10.7%) | 80 (11.1%) | 469 (12.2%) | 190 (12.0%) | 0.36 |
| COPD | 570 (32.3%) | 236 (32.7%) | 1678 (43.8%) | 675 (42.7%) | <0.001 |
| Cancer | 420 (23.8%) | 174 (24.1%) | 897 (23.4%) | 355 (22.5%) | 0.77 |

Continuous variables presented as mean (standard deviation) and analyzed using T-test. Dichotomous variables presented as frequency (percentage) and analyzed using Chi-square testing.

Abbreviations: BMI = body mass index; COPD = chronic obstructive pulmonary disease; CHF = congestive heart failure; mPAP = mean pulmonary artery pressure; RHC = right heart catheterization

*All comorbidities defined by Charlson comorbidity definitions, determined using ICD10 codes from VA claims.

### RHC and echocardiographic characteristics

RHC hemodynamics for analysis of *inpatient* and *outpatient cohorts* are presented in **Supplement 5** in S1 Data. In both analyses, mean right atrial pressure, pulmonary artery systolic and diastolic pressures, and PCWP were higher in patients with PH compared to those without PH. There were no significant differences in outpatient RHC hemodynamics between patients with or without outpatient COVID-19 positivity, both in patients with and without PH. Cardiac output and index, calculated by thermodilution, were also similar between the two groups.

Baseline echocardiographic parameters of the echocardiography cohort are shown in **Supplement 6** in S1 Data.

## Outcomes in inpatient and validation cohorts

The primary outcome of in-hospital mortality occurred in a total of 205 (17%) patients in the *inpatient cohort*. In-hospital mortality was 18% among those with mPAP > 20 mmHg and 13% for those with mPAP≤ 20mmHg. Within 30 days, those with mPAP > 20 mmHg had higher use of vasopressor medications (16.1% v. 10.1%), new dialysis (12.9% v. 7.1%), and low-flow oxygen use (79.1% v. 70.0%) compared to those with mPAP≤ 20mmHg (Table 3). Venous thromboembolism has been reported to be associated with COVID-19. We found that 7.9% of our inpatient cohort had VTE with 60 days of hospitalization, however the incidence of VTE was similar in the two groups (8.4% in mPAP > 20 mmHg group and 6.0% in mPAP ≤ 20 mmHg group, p=ns).

Multivariate logistic regression modeling showed that compared to patients without pre-existing PH, those with PH had higher in-hospital mortality risk (OR 1.60, 95% CI: 1.04–2.46). There was similarly significant association with in-hospital mortality when mPAP modeled as a continuous variable (OR 1.34, 95% 1.10–1.48, per 10 mmHg increase in mPAP). There was also a significant association between PH and 30-day low flow oxygen use (OR 1.46, 95% CI: 1.03–2.07). There was also a trend towards increased 30-day mortality, 30-day mechanical ventilation, 30-day vasopressor medication use, 30-day dialysis, and 30-day high-flow oxygen use with mPAP > 20 mmHg compared to mPAP ≤ 20 mmHg (Table 4). When we stratified the relationship between mPAP and in-hospital mortality by PCWP, in-hospital mortality risk was accentuated in patients with PCWP ≤ 15 mmHg (OR 1.93, 97.5% CI: 1.07–3.46) compared to PCWP > 15 (OR 1.24, 97.5% CI: 0.48–3.20) (**Supplement 7** in S1 Data).

In the *validation cohort,* there were 108 (16.5%) in-hospital deaths. There was higher mortality among patients with RVSP ≥ 30 mmHg compared to those with RVSP < 30 mmHg (20.5% vs 10.3%). After adjustment for multiple covariates, a significant increase in the risk of all-cause in-hospital mortality was observed with RVSP ≥ 30 mmHg (OR 2.12 95% CI 1.18–3.82) with a trend towards increased in-hospital mortality with RVSP modeled as a continuous variable (OR 1.10, 95% 0.90–1.22, per 10 mmHg increase in RVSP) (**Supplement 8** in S1 Data).

## Outcomes in outpatient cohort

In the outpatient cohort, the primary outcome of 1-year mortality occurred in a total of 407 (5.2%) patients. The event rates were highest for patients with a history of COVID-19 positive test with or without PH (8.98% and 7.07%, respectively) compared to patients with a negative COVID-19 test (4.41% with PH and 2.55% without PH). A total of 1,718 patients were hospitalized within 1 year (21.8%). Patients with PH had higher hospitalization rates (22–26%) compared to patients without PH (~18%). Event rates are presented in **Table 5**.

**Table 3. Event Rates for Inpatient Cohort.**

|  | mPAP ≤ 20 mmHg | mPAP > 20 mmHg | Total | p-value |
|---|---|---|---|---|
|  | N=252 | N=952 | N=1,204 |  |
| Mortality during COVID hospital stay | 33 (13.10%) | 172 (18.07%) | 205 (17.03%) | 0.06 |
| ICU during hospital stay | 27 (10.71%) | 84 (8.82%) | 111 (9.22%) | 0.36 |
| Mechanical Ventilation within 30 days | 22 (9.24%) | 124 (13.89%) | 146 (12.91%) | 0.06 |
| High Flow Oxygen within 30 days | 61 (25.63%) | 239 (26.76%) | 300 (26.53%) | 0.73 |
| Received high flow oxygen and/or mechanical ventilation within 30 days | 65 (25.79%) | 277 (29.10%) | 342 (28.41%) | 0.301 |
| Vasopressor Medication within 30 days | 24 (10.08%) | 144 (16.13%) | 168 (14.85%) | 0.02 |
| New Dialysis within 30 days | 17 (7.14%) | 115 (12.88%) | 132 (11.67%) | 0.01 |
| Low Flow Oxygen within 30 days | 166 (69.75%) | 706 (79.06%) | 872 (77.10%) | 0.002 |

Dichotomous variables presented as frequency (percentage) and analyzed using Chi-square testing.

Abbreviations: mPAP = mean pulmonary artery pressure

**Table 4. Relationship Between mPAP and Outcome in Inpatient Cohort in Multivariate Adjusted Model.**

| Outcome | Exposure | Odds Ratio | 95% CI |
|---|---|---|---|
| Mortality during hospital stay | mPAP > 20 vs. ≤ 20 | 1.60 | (1.04, 2.46) |
| | mPAP Continuous | 1.34 | (1.10, 1.48) |
| ICU during hospital stay | mPAP > 20 vs. ≤ 20 | 0.84 | (0.51, 1.43) |
| | mPAP Continuous | 1.00 | (0.82, 1.22) |
| Mortality within 30 days of admission | mPAP > 20 vs. ≤ 20 | 1.29 | (0.85, 1.97) |
| | mPAP Continuous | 1.10 | (0.90, 1.34) |
| Mechanical Ventilation within 30 days of admission | mPAP > 20 vs. ≤ 20 | 1.50 | (0.91, 2.49) |
| | mPAP Continuous | 1.10 | (0.90, 1.22) |
| Vasopressor Medication within 30 days of admission | mPAP > 20 vs. ≤ 20 | 1.49 | (0.92, 2.41) |
| | mPAP Continuous | 1.34 | (1.10, 1.59) |
| New Dialysis within 30 days of admission | mPAP > 20 vs. ≤ 20 | 1.70 | (0.96, 3.01) |
| | mPAP Continuous | 1.34 | (1.10, 1.63) |
| High Flow Oxygen within 30 days of admission | mPAP > 20 vs. ≤ 20 | 1.04 | (0.74, 1.47) |
| | mPAP Continuous | 1.00 | (0.90, 1.22) |
| Low Flow Oxygen within 30 days of admission | mPAP > 20 vs. ≤ 20 | 1.46 | (1.03, 2.07) |
| | mPAP Continuous | 1.48 | (1.22, 1.63) |
| Ventilator and/or High Flow Oxygen within 30 days of admission | mPAP > 20 vs. ≤ 20 | 1.16 | (0.83, 1.62) |
| | mPAP Continuous | 1.00 | (0.90, 1.22) |

Abbreviations: ICU = intensive care unit; mPAP = mean pulmonary artery pressure Logistic-regression adjusted for demographic age, sex, race, marital status, history of myocardial infarction, congestive heart failure, peripheral vascular disease, cerebrovascular accident or transient ischemic attack, dementia, chronic obstructive pulmonary disease, connective tissue disease, peptic ulcer disease, liver disease, diabetes mellitus, moderate to severe chronic kidney disease, solid tumor, leukemia, lymphoma, acquired immunodeficiency syndrome), and number of hospitalizations and VA emergency department visits in the one year preceding the positive COVID-19 test with time and region fixed effects, clustered at the patient level.

mPAP continuous modeled as per 10 mmHg increase in mPAP.

A total of 184 patients (2.3%) died within the first 90 days, and a total of 682 (8.6%) were hospitalized between 0–90 days of COVID-19 positivity. Of those who survived to day 91 (n=7,713), 223 (2.9%) died between days 91–365, and 1,081 (14%) were hospitalized between days 91–365. Event rates of death and hospitalization are presented in **Supplement 9** in S1 Data for 0–90 and 90–365 days. Kaplan-Meier curves for mortality stratified by the four groups are shown in **Fig 1a and 1b**, showing a predominant effect of outpatient COVID-19 positivity on mortality risk in the first 90 days and a comparable effect of COVID-19 and PH on mortality risk after the first 90 days. In the multivariable adjusted model, those with COVID-19 as an outpatient had significantly increased risk of 90-day mortality (hazard ratio ~8) irrespective of the presence of PH compared to patients who did not test positive for COVID-19 and did not have PH (**Table 6**). The 90-day hazard ratio of hospitalization alone and the combined outcome of mortality and hospitalization over 90 days was about three-to-four-fold irrespective of presence of PH. Those with PH alone (negative for COVID-19) did not have increased risk of 90-day mortality or hospitalization (Table 6**).**

In patients who survived to day 91, the mortality risk from 91 days to 1 year of follow up was significantly higher in all three groups compared to the no COVID-19 and no PH group. The risk of 1-year mortality in the outpatient setting beyond the first 90 days after a positive COVID-19 test, was comparable among the three groups, with hazard ratios ranging from 1.77 to 2.87, with largely overlapping confidence intervals (**Table 6**). After 90 days of outpatient COVID-19 positivity, the hazard for hospitalization was 24% higher only in patients who had both underlying PH and had tested positive for COVID-19 and not in any other groups, a trend that was similar for hazards of combined outcome of hospitalization and mortality. Findings were similar when assessing a subgroup of patients with PCWP ≤ 15 mmHg (**Supplement 10** in S1 Data).

**Table 5. Event Rates for Outpatient Cohort.**

| | Negative Outpatient COVID-19 Test, mPAP ≤ 20 mmHg | Positive Outpatient COVID-19 Test, mPAP ≤ 20 mmHg | Negative Outpatient COVID-19 Test, mPAP > 20 mmHg | Positive Outpatient COVID-19 Test, mPAP > 20 mmHg | p-value |
|---|---|---|---|---|---|
| | (n=1766) | (n=721) | (n=3829) | (n=1581) | |
| Died within 1 year | 45 (2.55%) | 51 (7.07%) | 169 (4.41%) | 142 (8.98%) | <0.001 |
| Hospitalized within 1 year | 326 (18.46%) | 132 (18.31%) | 849 (22.17%) | 411 (26.00%) | <0.001 |
| Death or Hospitalization within 1 year | 359 (20.33%) | 171 (23.72%) | 974 (25.44%) | 510 (32.26%) | <0.001 |

Abbreviations: mPAP = mean pulmonary artery pressure

### 0-90 Days

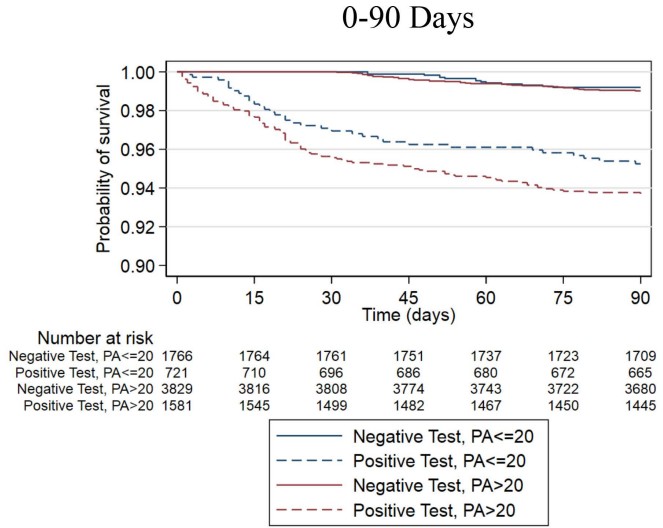

Kaplan-Meier curves comparing mortality rates from days 0-90 among Veterans with negative outpatient COVID-19 test and mPAP of ≤ 20 mmHg, positive outpatient COVID-19 test and a mPAP of ≤ 20 mmHg, negative outpatient COVID-19 test and a mPAP > 20 mmHg, and positive outpatient COVID-19 test and mPAP > 20 mmHg.

### 91-365 Days

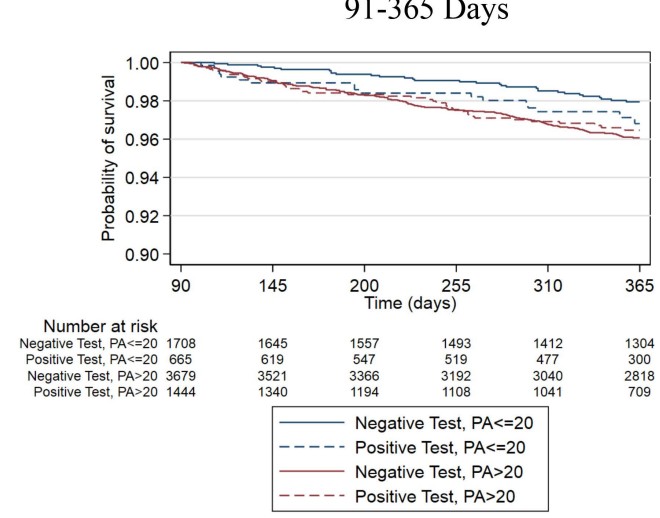

Kaplan-Meier curves comparing mortality rates from days 91-365 among Veterans with negative outpatient COVID-19 test and mPAP of ≤ 20 mmHg, positive outpatient COVID-19 test and a mPAP of ≤ 20 mmHg, negative outpatient COVID-19 test and a mPAP > 20 mmHg, and positive outpatient COVID-19 test and mPAP > 20 mmHg.

**Fig 1. Kaplan-Meier Survival Estimates.** Kaplan-Meier curves comparing mortality rates from days 0-90 (left) and days 90-365 (right) among Veterans with negative outpatient COVID-19 test and mPAP of ≤ 20 mmHg, positive outpatient COVID-19 test and a mPAP of ≤ 20 mmHg, negative outpatient COVID-19 test and a mPAP > 20 mmHg, and positive outpatient COVID-19 test and mPAP > 20 mmHg.

## Discussion

In a large, national cohort, our study demonstrates the influence of pre-existing PH on both acute outcomes following hospitalization for COVID-19 and longer-term outcomes for those who tested positive for COVID-19 in an outpatient setting. Pre-existing PH in patients hospitalized with COVID-19 increased risk of in-hospital mortality by 60% to 110% when PH was demonstrated on hemodynamics or echocardiography, respectively. In the outpatient setting, a heightened mortality risk appeared to be significantly influenced by COVID-19 positivity, with at least an 8-fold and a 2.8-fold increased risk of death in the first 90-days and 91–365 days, respectively, irrespective of presence of PH. Similarly, the risk of 90-day hospitalization was significantly increased among those COVID-19, irrespective of presence of PH.

**Table 6. Relationship Between COVID-19 Positivity, PH, and Mortality and Hospitalization Adjusted Outcomes in Outpatient Cohort.**

| | 0-90 days | | | 91-365 days | | |
| | n=7,897 | | | n=7,713 | | |
| | HR | 97.5% CI | p-value | HR | 97.5% CI | p-value |
|---|---|---|---|---|---|---|
| **Outcome: Mortality** | | | | | | |
| COVID-19 Positive, No PH vs. COVID-19 Negative, No PH | 8.23 | (3.96, 17.11) | <0.001 | 2.87 | (1.43, 5.75) | 0.001 |
| COVID-19 Negative, PH vs. COVID-19 Negative, No PH | 1.09 | (0.53, 2.22) | 0.793 | 1.77 | (1.13, 2.79) | 0.004 |
| COVID-19 Positive PH vs. COVID-19 Negative, No PH | 9.82 | (4.99, 19.31) | <0.001 | 2.85 | (1.64, 4.94) | <0.001 |
| **Outcome: Hospitalization** | | | | | | |
| COVID-19 Positive, No PH vs. COVID-19 Negative, No PH | 2.60 | (1.69, 4.00) | <0.001 | 0.84 | (0.60, 1.18) | 0.259 |
| COVID-19 Negative, PH vs. COVID-19 Negative, No PH | 0.87 | (0.61, 1.25) | 0.393 | 1.05 | (0.87, 1.25) | 0.578 |
| COVID-19 Positive, PH vs. COVID-19 Negative, No PH | 3.20 | (2.23, 4.60) | <0.001 | 1.24 | (0.99, 1.55) | 0.035 |
| **Outcome: Mortality or hospitalization** | | | | | | |
| COVID-19 Positive, No PH vs. COVID-19 Negative, No PH | 3.39 | (2.34, 4.91) | <0.001 | 0.97 | (0.65, 1.45) | 0.860 |
| COVID-19 Negative, PH vs. COVID-19 Negative, No PH | 0.91 | (0.66, 1.25) | 0.504 | 1.16 | (0.95, 1.43) | 0.102 |
| COVID-19 Positive, PH vs. COVID-19 Negative, No PH | 4.06 | (2.93, 5.61) | <0.001 | 1.50 | (1.15, 1.97) | 0.001 |

Cox-regression adjusted for demographic age, sex, race, marital status and history of myocardial infarction, congestive heart failure, peripheral vascular disease, cerebrovascular accident or transient ischemic attack, dementia, chronic obstructive pulmonary disease, connective tissue disease, peptic ulcer disease, liver disease, diabetes mellitus, moderate to severe chronic kidney disease, solid tumor, leukemia, lymphoma, acquired immunodeficiency syndrome, and number of hospitalizations and VA emergency department visits in the one year preceding the positive COVID-19 test with time and region fixed effects, clustered at the patient level.

In contrast, the risk of 1-year hospitalization beyond 90-days was significantly increased only in the group with PH and COVID-19.

To the best of our knowledge, our study is the first to assess the impact of pre-existing PH in patients admitted with COVID-19 on in-hospital outcomes and validated in a separate cohort, and to assess the long-term impact of both positive COVID-19 outpatient test and invasively defined PH. Previous case reports and small case series have been published describing *acute* PH [10], heart failure with cor pulmonale [6,15], and pulmonary embolism [7] in patients admitted with COVID-19 but none focused on preexisting PH or outpatient settings. While initially postulated that both the underlying pathophysiology and therapies specific to PAH may have a protective role against severe COVID-19 [16], a survey-based estimate of patients with PAH/CTEPH and COVID-19 found that 30% required hospitalization and 12% died, both of which are significantly greater than the general population [17]. Our observations strengthen this perspective by including more common forms of PH resulting from cardiac and pulmonary diseases in a large cohort of PH patients, adjusting for the effects of comorbidities, and using two distinct cohorts (one VA and another non-VA) and two different modalities to assess for PH (RHC and echocardiography), highlighting the robustness and generalizability of our results.

Patients with PH had elevated pulmonary capillary wedge pressure in both cohorts suggestive of left heart disease as the predominant etiology of PH. Similarly, PH patients had higher mitral valve E/e' ratio and left atrial volume indices consistent with left heart disease in addition to elevated right atrial pressures. However, the effect of PH on outcomes was independent of history of heart failure in either cohort, and mortality risk was accentuated in the setting of low PCWP in subgroup analyses, suggesting that the relationship between PH and poor outcomes is unlikely to be explained by left heart failure alone and perhaps reflective of right heart dysfunction.

The observed distinction in the effect of pre-existing PH on mortality risk in inpatient vs. outpatient settings was surprising and a pivotal observation. The lack of synergistic increase in mortality risk with PH in the outpatient settings may be due to the differential disease dynamics between the inpatient and outpatient settings. In severe COVID-19 infection requiring hospitalization, there may be significant hypoxia, which worsens PH, and thrombotic complications, which further increase RV afterload. Thus, the synergistic effects between pre-existing PH and inpatient COVID-19 outcomes could be caused by hypoxia and vascular dysfunction exacerbating RV strain and leading to worse outcomes. In the outpatient setting, despite lack of hypoxia or significant thrombotic complications, there is still a far greater risk of overall mortality in the first 90 days up until 1 year with outpatient COVID-19 positivity compared to outpatient COVID-19 negativity. In the outpatient cohort, there was a 5.2% rate of 1-year mortality, which is far greater than previous large, population studies showing mortality rates of 0.2% with COVID-19 managed in the outpatient setting [18]. Our cohort of patients who received prior RHC may be inherently a higher risk population with pre-existing comorbidity burden. Thus, the absence of synergistic effect might be due to the predominant effects of COVID-19, particularly within the initial 90 days, which may overshadow the impact of pre-existing PH in this cohort. As follow-up extends beyond 90 days, mortality risk in patients with PH compared to those without, even in the absence of COVID-19 positivity, increases by 68%, highlighting that the chronic nature of PH becomes more influential in the longer term.

Interestingly, while the risk of hospitalization varied between the two time periods showing the effect of COVID-19 in first 90 days, this hospitalization risk only persisted in the group with PH after 90 days. These findings also persisted when hospitalization and mortality were combined, excluding mortality as a competing risk. Considering sample size, the outcome of hospitalization between 91–365 days occurred most in patients with PH and negative outpatient COVID-19 test (572 patients), a two-fold increase from the outcome of hospitalization within 90 days in patients with PH and negative outpatient COVID-19 test (284 patients), which may have driven these findings. The difference between hospitalization findings and mortality may also be explained by varying healthcare utilization patterns in the first 90 days compared to 91–365 days.

The lasting impact and sequelae of COVID-19 infection and the possibility of future surges are unknown. Thus, there is an urgent need to identify populations at highest risk of adverse outcomes from COVID-19 infection such as those with pre-existing PH in order to identify ways to mitigate this risk. Also, while there is increasing recognition of post-acute sequelae of COVID-19, especially in patients who required hospitalization due to COVID-19 illness, the long-term impacts of COVID-19 managed in the outpatient setting are less clear. This is important, as the vast majority of people with COVID-19 were not hospitalized. The results of our study provide a timeframe of when patients are at highest risk of acute COVID-19 related outcomes versus long-term PH related outcomes. This nuanced understanding of PH and COVID-19 related outcomes facilitates risk stratification and closer monitoring for adverse outcomes. Future studies are warranted to further define this at-risk population and prospectively assess longer-term outcomes in patients with PH and COVID-19 infection.

There are several important limitations to our study. First, this was a retrospective observational study, and we cannot rule out the possibility of residual confounding or bias despite the use of multivariable adjustment. Second, the use of Charlson comorbidity definitions, determined using ICD-10 codes, to define clinical outcomes may have resulted in under- or mis-classification; however, had this occurred, it would have biased our results toward the null. Third, PH is a heterogenous group of disorders with complex patient profiles and invasive hemodynamics. It is possible that our results would have been more or less pronounced in specific World Symposium on Pulmonary Hypertension groups, such as those with left heart failure. Also, pre-admission echocardiograms or RHC showing elevated PA and/or filling pressures may have resulted in patient treatment with therapies such as diuretics or pulmonary vasodilators prior to development of COVID-19; however, this would have also biased our results toward the null. Fourth, while the study spanned multiple COVID-19 variants, including the Omicron era, we did not have data on the variant type or vaccination status for individual patients. However, after analysis, neither the COVID-19 dates nor vaccination status were found to be significant predictors of

outcomes in our cohort. Fifth, since our analysis was based on ICD-coded data, we were unable to assess granular clinical details related to the severity of illness. Lastly, we do not have information on cause of death which limits our ability to precisely evaluate the pathophysiological link between our exposures and mortality. Future studies incorporating comprehensive clinical and imaging data may provide deeper insights into the mechanisms underlying the observed associations and further clarify causes of death in this population.

## Conclusions

Patients with pre-admission PH admitted with COVID-19 had higher rates of in-hospital mortality than those without pre-admission PH, both in a national VA cohort and validated in a local hospital network. When assessing longer-term outcomes, mortality risk seemed to predominantly be influenced by COVID-19 positivity in the outpatient setting irrespective of PH, while hospitalization risk persisted only in patients with PH after 90 days.

## Supporting information

**S1 Data.  PH Manuscript Supplemental Data.**
(DOCX)

## Acknowledgements

Drs. Vijayakumar, Louis, Erqou, Wu, and Choudhary contributed to conception and design. Ms. Corneau and Mr. Has performed statistical analyses. Drs. Vijayakumar, Louis, Erqou, Waldo, Plomondon, Sheikh, Saad, Abbott, Jankowich, Aronow, Wu, and Choudhary, and Ms. Gokhale contributed to acquisition, analysis, or interpretation of data, drafting of the manuscript, or reviewing the manuscript for important intellectual content. We thank Matthew Borgia for his help with data analyses.

## Author contributions

**Conceptualization:** Herbert D. Aronow, Wen-Chih Wu, Gaurav Choudhary.

**Data curation:** Herbert D. Aronow, Wen-Chih Wu, Gaurav Choudhary.

**Formal analysis:** David W. Louis, Emily Corneau, Phinnara Has, Gaurav Choudhary.

**Funding acquisition:** Gaurav Choudhary.

**Investigation:** Sebhat Erqou, Wen-Chih Wu, Gaurav Choudhary.

**Methodology:** Shilpa Vijayakumar, Sebhat Erqou, Herbert D. Aronow, Wen-Chih Wu, Gaurav Choudhary.

**Project administration:** Gaurav Choudhary.

**Resources:** Herbert D. Aronow, Gaurav Choudhary.

**Software:** Gaurav Choudhary.

**Supervision:** Wen-Chih Wu, Gaurav Choudhary.

**Validation:** Gaurav Choudhary.

**Visualization:** Wen-Chih Wu, Gaurav Choudhary.

**Writing – original draft:** Shilpa Vijayakumar, David W. Louis, Sebhat Erqou, Wasiq Sheikh, Herbert D. Aronow, Wen-Chih Wu, Gaurav Choudhary.

**Writing – review & editing:** David W. Louis, Stephen W. Waldo, Mary E. Plomondon, Madhura Gokhale, Marwan Saad, J. Dawn Abbott, Matthew Jankowich, Herbert D. Aronow, Wen-Chih Wu, Gaurav Choudhary.

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
