## [Decision Letter · Decision Letter 0]

4 Dec 2024

PONE-D-24-34936Association Between Pulmonary Hypertension and COVID-19 Related Outcomes in Inpatient and Ambulatory Care SettingsPLOS ONE

Dear Dr. Choudhary,

Thank you for submitting your manuscript to PLOS ONE. After careful consideration, we feel that it has merit but does not fully meet PLOS ONE’s publication criteria as it currently stands. Therefore, we invite you to submit a revised version of the manuscript that addresses the points raised during the review process.

We look forward to receiving your revised manuscript.

Kind regards,

Fateen Ata, MD

Academic Editor

PLOS ONE

Journal Requirements:

2. We note that your Data Availability Statement is currently as follows: “All relevant data are within the manuscript and in Supporting Information files.”

Please confirm at this time whether or not your submission contains all raw data required to replicate the results of your study. Authors must share the “minimal data set” for their submission. PLOS defines the minimal data set to consist of the data required to replicate all study findings reported in the article, as well as related metadata and methods (https://journals.plos.org/plosone/s/data-availability#loc-minimal-data-set-definition). For example, authors should submit the following data: - The values behind the means, standard deviations and other measures reported; - The values used to build graphs; - The points extracted from images for analysis. Authors do not need to submit their entire data set if only a portion of the data was used in the reported study. If your submission does not contain these data, please either upload them as Supporting Information files or deposit them to a stable, public repository and provide us with the relevant URLs, DOIs, or accession numbers. For a list of recommended repositories, please see https://journals.plos.org/plosone/s/recommended-repositories. If there are ethical or legal restrictions on sharing a de-identified data set, please explain them in detail (e.g., data contain potentially sensitive information, data are owned by a third-party organization, etc.) and who has imposed them (e.g., an ethics committee). Please also provide contact information for a data access committee, ethics committee, or other institutional body to which data requests may be sent. If data are owned by a third party, please indicate how others may request data access.

3. Please remove your figures from within your manuscript file, leaving only the individual TIFF/EPS image files, uploaded separately. These will be automatically included in the reviewers’ PDF**.**

Additional Editor Comments:

We have completed the peer-review process, and I'm writing to inform you that your submission has been evaluated as requiring revisions.

The reviewers have provided constructive feedback and identified key areas that need substantial revision. Their comments aim to enhance the scientific rigor, clarity, and impact of your work. We believe your research will benefit from a revision, and we encourage you to revise the manuscript per the comments provided.

Please find the reviewers' comments attached.

Reviewers' comments:

Reviewer's Responses to Questions

**Comments to the Author**

1. Is the manuscript technically sound, and do the data support the conclusions?

Reviewer #1: Yes

Reviewer #2: Yes

Reviewer #3: Partly

2. Has the statistical analysis been performed appropriately and rigorously? 

Reviewer #1: Yes

Reviewer #2: Yes

Reviewer #3: Yes

3. Have the authors made all data underlying the findings in their manuscript fully available?

Reviewer #1: Yes

Reviewer #2: Yes

Reviewer #3: Yes

4. Is the manuscript presented in an intelligible fashion and written in standard English?

Reviewer #1: Yes

Reviewer #2: Yes

Reviewer #3: Yes

5. Review Comments to the Author

Reviewer #1: Dear Authors, this is a well designed and excellent study. As we know from new studies COVID-19 patients may have pulmonary embolism during the illness and PE can cause to Pulmonary hypertension. Is it possible to include your PE data of the study groups.

Sincerely.

Reviewer #2: This article offers insight into the association of pre-existing PH to the outcome of COVID-19. The concept is innovative, and the methods used are sound. Overall, the article is of sufficient quality for publication, though I have a few minor questions and suggestions.

1. I would suggest adding “pre-existing” in the title, to emphasize you are focusing on pre-existing PH in this article, because there are plenty of papers about PH with COVID and “pre-existing” is the major novelty in this study.

2. Table 1 actually shows that HP patients are younger, moreover, they also have higher BMI, which does not fit the narrative in the result section that PH patients are older.

3. Table 3, incidence of mechanical ventilation is not significant (p=0.06), you might need to tune down a little in the corresponding result session.

4. In discussion, “Pre-existing PH in patients hospitalized with COVID-19 increased risk of in-hospital mortality by 60% to 110%”, where do the numbers come from?

Reviewer #3: The study is interesting as it aims to analyse the impact of pre existing PH in Covid 19 patients amongst patients in both the inpatient and the outpatient settings.However, I have certain questions for the authors which I believe they would clarify.

1) How was Covid 19 diagnosed in the patients? Was an RTPCR or RAT used? What was the radiological pattern in these patients? Was HRCT done and was there a correlation?

2) Was the presence of fever, high serum ferritin or D dimer values taken into account while admitting the patients? In short what were the criteria for admission and was there a correlation between these factors?

3) Were cardiac enzymes done in the patients and could this be a confounding factor?

4 ) The authors have not mentioned what drugs or therapies the patients were on for their PH. Certain studies have shown patho physiological features of pulmonary hypertension and targeted therapies can lead to a protective effect in Covid 19 due to decreased viral entrance through reduced ACE -2 expression, an attenuated response to lung perfusion changes and vasodilator therapy may all have an impact in the outcome of the study.

5) The authors have mentioned some of the comorbidities but were there patients of OSA in the study group. Also was the BMI of the patients known because again there are studies to suggest that frailty might be a better indicator of outcome.

6) How many patients were included in the Omicron era. Do the authors believe that this could have been a confounding factor as also vaccination. I understand that this is a retrospective study but, lot of questions need to be answered.

Thank you.

6. PLOS authors have the option to publish the peer review history of their article (what does this mean? ). If published, this will include your full peer review and any attached files.

**Do you want your identity to be public for this peer review?** For information about this choice, including consent withdrawal, please see our Privacy Policy .

Reviewer #1: **Yes: ** ERSIN DEMIRER

Reviewer #2: No

Reviewer #3: No

---

## [Author Response · Author response to Decision Letter 1]

28 Feb 2025

Response to Reviews:

Reviewer #1:

1) Dear Authors, this is a well-designed and excellent study. As we know from new studies COVID-19 patients may have pulmonary embolism during the illness and PE can cause to Pulmonary hypertension. Is it possible to include your PE data of the study groups.

We sincerely thank the reviewer for their insightful feedback. We agree that pulmonary embolism (PE) is an important consideration in the context of COVID-19. Using ICD-codes for venous thromboembolism (VTE) within 60 days post-index hospitalization, we found that a total of 95 out of 1204 patients (7.9%) in our inpatient cohort had VTE, of which 80 (8.4%) had mPAP > 20 mmHg and 15 (6.0%) had mPAP < 20 mmHg. There was no significant difference in the incidence of VTE between the two groups.

Results Page 9: “Venous thromboembolism has been reported to be associated with COVID-19. We found that 7.9% of our inpatient cohort had VTE with 60 days of hospitalization, however the incidence of VTE was similar in the two groups ( 8.4% in mPAP > 20 mmHg group and 6.0% in mPAP ≤ 20 mmHg group, p=ns).”

Reviewer #2:

This article offers insight into the association of pre-existing PH to the outcome of COVID-19. The concept is innovative, and the methods used are sound. Overall, the article is of sufficient quality for publication, though I have a few minor questions and suggestions.

1) I would suggest adding “pre-existing” in the title, to emphasize you are focusing on pre-existing PH in this article, because there are plenty of papers about PH with COVID and “pre-existing” is the major novelty in this study.

We sincerely thank the reviewer for their thoughtful feedback. We appreciate the recognition of our study’s innovation and methodological rigor. Based on the reviewer’s valuable suggestion, we have revised the title to explicitly include “pre-existing” to better highlight the focus of our study. We believe this change enhances clarity.

2) Table 1 actually shows that PH patients are younger, moreover, they also have higher BMI, which does not fit the narrative in the result section that PH patients are older.

We thank the reviewer for this observation. We have now revised the text to accurately reflect our findings, clarifying that in our cohort, patients with pre-existing PH were younger and had a higher BMI, as quoted below. We appreciate the reviewer’s attention to detail.

Results, page 8: “Compared to those without PH, patients with PH were younger, had higher body mass index (BMI), older and were more likely to have comorbidities such as congestive heart failure (CHF), diabetes mellitus, and chronic obstructive pulmonary disease (COPD).”

3) Table 3, incidence of mechanical ventilation is not significant (p=0.06), you might need to tune down a little in the corresponding result session.

We thank the reviewer for this clarification. To avoid overstatement, we have adjusted the corresponding section in the results to reflect this appropriately, now only stating that there is a non-significant trend towards increased 30-day mechanical ventilation with mPAP > 20 mmHg compared to mPAP � 20 mmHg.

Results, page 9: “Within 30 days, those with mPAP > 20 mmHg had higher use of vasopressor medications (16.1% v. 10.1%), new dialysis (12.9% v. 7.1%), and low-flow oxygen use (79.1% v. 70.0%) compared to those with mPAP� 20mmHg (Table 3).

Multivariate logistic regression modeling showed that compared to patients without pre-existing PH, those with PH had higher in-hospital mortality risk (OR 1.60, 95% CI: 1.04-2.46). There was similarly significant association with in-hospital mortality when mPAP modeled as a continuous variable (OR 1.34, 95% 1.10-1.48, per 10 mmHg increase in mPAP). There was also a significant association between PH and 30-day low flow oxygen use (OR 1.46, 95% CI: 1.03-2.07). There was also a non-significant trend towards increased 30-day mortality, 30-day mechanical ventilation, 30-day vasopressor medication use, 30-day dialysis, and 30-day high-flow oxygen use with mPAP > 20 mmHg compared to mPAP � 20 mmHg (Table 4).”

4) In discussion, “Pre-existing PH in patients hospitalized with COVID-19 increased risk of in-hospital mortality by 60% to 110%”, where do the numbers come from?

The reported increase in in-hospital mortality risk (60%; 110%) is derived from the odds ratios presented in our study. Specifically, the odds ratio for mortality in patients with mPAP > 20 mmHg versus < 20 mmHg was 1.60 (table 4), indicating a 60% increased risk, and the odds ratio for mortality in patients with RVSP > 30 mmHg versus < 30 mmHg was 2.12 (supplement 7), indicating a 112% increased risk, rounded to 110%.

Reviewer #3: The study is interesting as it aims to analyse the impact of pre existing PH in Covid 19 patients amongst patients in both the inpatient and the outpatient settings. However, I have certain questions for the authors which I believe they would clarify.

1) How was Covid 19 diagnosed in the patients? Was an RTPCR or RAT used? What was the radiological pattern in these patients? Was HRCT done and was there a correlation?

We thank the reviewer for their thoughtful feedback. COVID-19 was diagnosed in our study population using either PCR or rapid antigen testing. Unfortunately, we did not have access to radiology data, including HRCT findings, as our study relied on ICD codes for data extraction. Therefore, we were unable to assess specific radiological patterns or correlations.

2) Was the presence of fever, high serum ferritin or D dimer values taken into account while admitting the patients? In short what were the criteria for admission and was there a correlation between these factors?

3) Were cardiac enzymes done in the patients and could this be a confounding factor?

We thank the reviewer for this important question. As our study was based on database analysis, we did not have access to detailed clinical parameters such as fever, serum ferritin, D-dimer levels or cardiac enzymes at the time of admission. Therefore, we were unable to assess their role in admission criteria or potential correlations with outcomes. We acknowledge this as a limitation of our study and have now explicitly mentioned it in the discussion. Future studies with more granular clinical data can further explore these factors and their relationship to outcomes, including causes of death.

Discussion, page 15: “Fifth, since our analysis was based on ICD-coded data, we were unable to assess granular clinical details related to the severity of illness. Lastly, we do not have information on cause of death which limits our ability to precisely evaluate the pathophysiological link between our exposures and mortality. Future studies incorporating comprehensive clinical and imaging data may provide deeper insights into the mechanisms underlying the observed associations and further clarify causes of death in this population.

4) The authors have not mentioned what drugs or therapies the patients were on for their PH. Certain studies have shown patho physiological features of pulmonary hypertension and targeted therapies can lead to a protective effect in Covid 19 due to decreased viral entrance through reduced ACE -2 expression, an attenuated response to lung perfusion changes and vasodilator therapy may all have an impact in the outcome of the study.

We have now presented relevant baseline medications used prior to index hospitalization in the supplementary material (Supplement 3). Notably, for ACE inhibitors, we found that a total of 511 out of 1204 patients (42%) were taking ACE inhibitors at baseline, of which 401 (42%) had mPAP > 20 mmHg and 110 (44%) had mPAP < 20 mmHg. There was no significant difference in usage between the two groups. However, we unfortunately do not have data on medication changes during hospitalization.

5) The authors have mentioned some of the comorbidities but were there patients of OSA in the study group. Also was the BMI of the patients known because again there are studies to suggest that frailty might be a better indicator of outcome.

BMI data is available in our study, and we interestingly found that patients with pre-existing pulmonary hypertension had a higher BMI compared to those without (mean BMI 30.3 for mPAP > 20 mmHg compared to mean BMI 28.7 for mPAP < 20 mmHg). Regarding, OSA, we found that a higher proportion of patients with pre-existing pulmonary hypertension had baseline OSA compared to those without pre-existing pulmonary hypertension. We have highlighted both of these findings in our results section.

Results, page 8: “Compared to those without PH, patients with PH were younger, had higher body mass index (BMI), older and were more likely to have comorbidities such as congestive heart failure (CHF), diabetes mellitus, obstructive sleep apnea (OSA), and chronic obstructive pulmonary disease (COPD).”

6) How many patients were included in the Omicron era. Do the authors believe that this could have been a confounding factor as also vaccination. I understand that this is a retrospective study but, lot of questions need to be answered.

We thank the reviewer for these insightful questions. We can infer the inclusion of patients in the Omicron era based on the dates of COVID diagnosis in our cohort (9/1/2021-2/28/2022), which included a total of 379 patients. While we acknowledge that the Omicron variant and vaccination status could potentially act as confounding factors, we found that neither COVID variant nor vaccination status significantly predicted the outcomes in our study. We have added a sentence to the discussion to clarify that these factors were not predictors of outcomes in our cohort.

Discussion, page 15: “Fourth, while the study spanned multiple COVID-19 variants, including the Omicron era, we did not have data on the variant type or vaccination status for individual patients. However, after analysis, neither the COVID-19 dates nor vaccination status were found to be significant predictors of outcomes in our cohort.”

We sincerely thank the reviewers for their thoughtful and constructive feedback. Their insights have significantly enhanced the quality of our manuscript, and we greatly appreciate their contributions. We thank you again for your continued consideration, and we hope you find our re-submitted manuscript suitable for publication.

---

## [Decision Letter · Decision Letter 1]

14 Mar 2025

Association Between Pre-Existing Pulmonary Hypertension and COVID-19 Related Outcomes in Inpatient and Ambulatory Care Settings

PONE-D-24-34936R1

Dear Dr. Choudhary,

We’re pleased to inform you that your manuscript has been judged scientifically suitable for publication and will be formally accepted for publication once it meets all outstanding technical requirements.

Kind regards,

Fateen Ata, MD

Academic Editor

PLOS ONE

Additional Editor Comments (optional):

Reviewers' comments:

Reviewer's Responses to Questions

**Comments to the Author**

1. If the authors have adequately addressed your comments raised in a previous round of review and you feel that this manuscript is now acceptable for publication, you may indicate that here to bypass the “Comments to the Author” section, enter your conflict of interest statement in the “Confidential to Editor” section, and submit your "Accept" recommendation.

Reviewer #1: All comments have been addressed

Reviewer #2: All comments have been addressed

Reviewer #3: All comments have been addressed

2. Is the manuscript technically sound, and do the data support the conclusions?

Reviewer #1: Yes

Reviewer #2: Yes

Reviewer #3: Yes

3. Has the statistical analysis been performed appropriately and rigorously? 

Reviewer #1: Yes

Reviewer #2: Yes

Reviewer #3: Yes

4. Have the authors made all data underlying the findings in their manuscript fully available?

Reviewer #1: Yes

Reviewer #2: Yes

Reviewer #3: Yes

5. Is the manuscript presented in an intelligible fashion and written in standard English?

Reviewer #1: Yes

Reviewer #2: Yes

Reviewer #3: Yes

6. Review Comments to the Author

Reviewer #1: Dear Authors, the VTE data has been presented. İ recommend your manuscript is worth being publishing to the Editorial team.

Sincerely.

Reviewer #2: The authors have answered my questions and addressed my concerns, and I believe the revised manuscript is good for publication.

Reviewer #3: The authors have addressed all the queries to satisfaction .They have also discussed the same in the discussion.The study has now become crisp and more clear. In future, they could do more studies so that the limitations of this study are addressed.The article may be considered for publication if it meets the other reviewers and editor’s criteria for publication.

7. PLOS authors have the option to publish the peer review history of their article (what does this mean? ). If published, this will include your full peer review and any attached files.

**Do you want your identity to be public for this peer review?** For information about this choice, including consent withdrawal, please see our Privacy Policy .

Reviewer #1: **Yes: ** Ersin Demirer

Reviewer #2: No

Reviewer #3: No

---

## [Editor Report · Acceptance letter]

PONE-D-24-34936R1

PLOS ONE

Dear Dr. Choudhary,

I'm pleased to inform you that your manuscript has been deemed suitable for publication in PLOS ONE. Congratulations! Your manuscript is now being handed over to our production team.

Kind regards,

on behalf of

Dr. Fateen Ata

Academic Editor

PLOS ONE